# Asphalt Binder Modification with Plastomeric Compounds Containing Recycled Plastics and Graphene

**DOI:** 10.3390/ma15020516

**Published:** 2022-01-10

**Authors:** Simone D’Angelo, Gilda Ferrotti, Fabrizio Cardone, Francesco Canestrari

**Affiliations:** Department of Civil and Building Engineering and Architecture, Università Politecnica delle Marche, via Brecce Bianche, 60131 Ancona, Italy; g.ferrotti@univpm.it (G.F.); f.cardone@univpm.it (F.C.); f.canestrari@univpm.it (F.C.)

**Keywords:** polymeric compounds, graphene, recycled plastics, rheological testing, bond strength test

## Abstract

Polymer-modified bitumens are usually employed for enhancing the mixture performance against typical pavement distresses. This paper presents an experimental investigation of bitumens added with two plastomeric compounds, containing recycled plastics and graphene, typically used for asphalt concrete dry modification. The goal was to study the effects of the compounds on the rheological response of the binder phase, as well the adhesion properties, in comparison with a reference plain bitumen. The blends (combination of bitumen and compounds) were evaluated through dynamic viscosity tests, frequency sweep tests, and multiple stress creep recovery (MSCR) tests. Moreover, the bitumen bond strength (BBS) test was performed to investigate the behavior of the systems consisting of blends and aggregate substrates (virgin and pre-coated). The rheological tests indicated that both blends performed better than the plain bitumen, especially at high temperature, showing an enhanced rutting resistance. In terms of bond strength, comparable results were found between the blends and reference bitumen. Moreover, no performance differences were detected between the two types of blends.

## 1. Introduction

It is well known that polymer-modified mixtures can be successfully employed for limiting road pavement distresses such as rutting, fatigue, and thermal cracking [1,2]. For the bituminous material modification, the dry or the wet method can be used. In the dry method, the polymers (generally in the form of pellets) are added to the asphalt mixture directly in the asphalt plant, leading to a modified asphalt concrete, whereas in the wet method, the polymers are added to the binder to obtain a polymer-modified bitumen (PMB), which is then added to the aggregates in the asphalt plant. In general, the wet method allows better control of the properties of the bitumen–polymer blend with respect to the dry process [3], even though it requires specific equipment to facilitate the mixing of the base bitumen with a selected modifying agent, as well as a proper handling during bitumen storage. The final characteristics of the PMB depend on the polymer content and on the compatibility between the polymer and base bitumen [4]. In fact, good performance is guaranteed if the two phases (a polymer-rich phase and an asphaltene-rich phase), which characterize the PMB, assure a good stability, i.e., a low phase separation [5], which depends on both the characteristics of bitumen and polymer, such as density, molecular weight, and solubility [6].

The two main categories of polymers currently used to modify bitumen are elastomers, such as styrene-butadiene-styrene (SBS) and styrene-butadiene-rubber (SBR), and plastomers, such as ethylene-vinyl-acetate (EVA), polyethylene (PE), and polypropylene (PP). Elastomers can significantly stretch under load and recover their initial shape when the load is removed, allowing for more flexible road pavements [7,8]. On the other hand, plastomers are characterized by a rigid three-dimensional network, resistant under load, but which can break. They increase pavement stiffness and improve resistance to permanent deformation [9,10,11,12].

Another promising category of modifying agents for bitumen is represented by graphene [13,14], able to improve the bitumen stiffness at high temperatures and the deformation recovery [15], as well as to reduce the cost of the entire pavement life cycle [16,17,18].

In addition, in recent decades, sustainable materials such as wastes from different origins have been investigated to be employed as modifying agents in asphalt pavements in order to promote the circular economy. Bitumens modified with crumb rubber from discarded tires, and their use for road pavements, have proved to be a valid alternative to the use of SBS [19,20,21] by providing several advantages to pavement performance, especially at high temperature. A recent challenge for the reduction of the environmental impact of plastics is represented by the possibility of using waste plastic to produce modified asphalt concretes [22], with both wet and dry processes. It has been shown that modified bitumens (wet method) produced with several types of waste plastic guarantee good physical and rheological properties, especially at high temperatures, provided that an optimum amount is added [23,24,25]. However, as mentioned above, the main aspect to consider in the wet procedure is the compatibility between bitumen and plastic, which depends on the characteristics of the materials. In general, recycled plastic can have a homogeneous dispersion in the bitumen but a poor storage stability over time [26,27]. On the other hand, modified asphalt mixtures obtained with the dry addition of plastics showed very good performance in terms of resistance to permanent deformations [28,29,30,31,32].

However, the modifying agents used for the asphalt concrete modification with the dry process are usually different from that used for wet modification, as they produce different effects on the material. In this sense, this research work has focused on simulating the chemical or physical modification produced on the binder phase by compounds used for dry modification, as well as comparing the performance of two different types of plastomeric compounds (recycled and not).

## 2. Objective

This study is part of a wider survey, aimed at investigating asphalt mixtures modified at the asphalt plant through the dry process, by using polymeric compounds, one of which contains recycled plastics and graphene.

As it is well known that the rheological characteristics of the mixture strongly depend on the properties of the constituting bitumen, in this initial study, two plastomeric compounds usually used for dry modification were grinded and mixed with a plain bitumen to obtain bituminous blends. Their effects on the rheological response of the binder phase, as well as the adhesion properties of the blends, were investigated.

Specifically, this experimental investigation was organized in different phases. Preliminarily, chemical analyses, such as Fourier Transform Infrared Spectroscopy (FTIR), Raman spectroscopy, Thermogravimetric Analysis (TGA), and Differential Thermal Analysis (DTA), were carried out on both the selected plastomeric compounds and corresponding bituminous blends produced. Later, the rheological properties of the blends were evaluated through dynamic viscosity tests, frequency sweep tests, and multiple stress creep recovery (MSCR) tests by using a Dynamic Shear Rheometer (DSR). Finally, the bitumen bond strength (BBS) test was performed to investigate the adhesion properties of the systems consisting of binder blends and aggregates (virgin and artificial). All the results were compared with those obtained for the reference condition represented by the plain bitumen.

## 3. Materials

### 3.1. Plain Bitumen

The main characteristics of the reference plain bitumen, classified as 50/70 pen grade (according to the EN 12591 [33]), are listed in Table 1.

### 3.2. Plastomeric Compounds

Two different types of compounds, coded as PC and GC, were added to the plain bitumen. PC is a gray-colored blend of plastomeric polymers, whereas GC is a dark-colored blend made with selected “rigid” recycled plastics and nonrecycled graphene. Both blends have a particularly hard (not soft) consistency and have the shape of pellets with a diameter less than 5 mm (Figure 1a,b). Their main properties are shown in Table 2.

Moreover, in order to investigate the characteristics of the two compounds more in depth, several preliminary chemical analyses were carried out on both the PC and GC compounds such as Fourier Transform Infrared Spectroscopy (FTIR), Raman spectroscopy, Thermogravimetric Analysis (TGA), and Differential Thermal Analysis (DTA).

#### 3.2.1. Fourier Transform Infrared Spectroscopy

FTIR analysis was conducted with a spectrometer, by using the U-ATR (Attenuated Total Reflectance) accessory for the reflection analysis. The spectrum of each granule of compound was acquired after 16 scans, performed at room temperature with a spectral resolution of 4 cm^−1^. For each compound, ten pellets were investigated.

Figure 2 shows that both compounds present all the bands attributable to the stretching, bending, and rocking of CH_2_ (2914, 2847, 1463, and 720 cm^−1^), the bands related to CH_3_ around 2900 cm^−1^, and the largest band at 1376 cm^−1^. They are very similar and basically consist of Polyethylene (PE) and Polypropylene (PP). In fact, the spectra of PE and PP (both available in the literature), reported for comparison with the GC spectrum in the inset of Figure 2, show that GC is substantially obtained by superimposing the spectra of PP and PE. Moreover, the PC compound is characterized by the bands related to oxidation, the most significant of which is located at 1730 cm^−1^ (carbonyl groups C=O).

#### 3.2.2. Raman Spectroscopy

One of the main objectives of the Raman Spectroscopy analysis is the identification of the presence of carbonous materials, such as graphene and graphene oxide, in the sample. In this case, the representative spectra of the two compounds (PC and GC) were examined in the most significant spectral range (800–1800 cm^−1^) to search for the two bands related to graphene:(1)The G-band, index of the lamellar structure of graphene oxide (stretching sp^2^);(2)The D-band, oscillation of the hexagonal carbon chains that occurs when the symmetry, linked to the functional groups and defects present on the plane and on the edge of the graphene oxide (GO), is broken.

The Raman Spectroscopy analysis was carried out on both the compounds through a micro-spectrometer, by using a 785 nm diode laser and a diffraction grating of 1200 lines/mm. A 10× lens was employed to inspect the compound pellets both in the surface and at depths of 100 and 200 microns, allowing the acquisition of point spectra.

The results showed that the sample is homogeneous as analog spectra were acquired on both the surface and in depth. Moreover, the representative average spectra (Figure 3) showed that the bands attributable to the absorptions relative to the G-band (1580 cm^−1^) and the D-band (1350 cm^−1^), characteristics of the presence of graphene, are present in modest quantities in the GC but absent in the PC compound.

#### 3.2.3. Thermogravimetric Analysis and Differential Thermal Analysis

The TGA consists of measuring the mass variation of a sample of material as a function of temperature under controlled atmosphere conditions and is mainly used to study the composition of the material and its thermal stability. The main result of this analysis is the thermogravimetric profile given by the variation in the mass of the sample (TG) and its first derivative (DTG) as a function of the temperature. The DTA consists of measuring the temperature difference between the analyzed sample and an inert reference material as the temperature varies over time. The main result of DTA analysis is the differential thermal profile that shows the temperature difference as a function of the temperature. TGA and DTA analyses were carried out on both PC and GC compounds, by considering a sample of about 25 mg and alumina crucibles as inert support. The materials were heated in nitrogen by increasing the temperature from 25 °C to 600 °C with an increase rate of 20 °C/min.

The results obtained for the PC compound are shown in Figure 4. The thermogravimetric analysis (Figure 4a) highlighted an overall mass loss of 96% with the onset temperature placed at 455 °C and the maximum degradation rate at 481 °C. The differential thermal profile DTA (Figure 4b) showed the presence of two endothermic peaks (at 127 and 161 °C), which are consistent with the presence of a copolymer containing ethylene and propylene and are due to the melting of the polymer crystalline parts.

The TGA analysis conducted on the GC compound (Figure 5a) showed an overall mass loss of 95% with the onset temperature placed at 462 °C and the maximum degradation rate at 489 °C. The DTA (Figure 5b) highlighted the presence of two endothermic peaks located at 141 and 167 °C, due to the melting of the polymer crystalline parts. Additionally in this case, the melting temperature is consistent with the presence of a copolymer containing ethylene and propylene. On the basis of these results and considering the similar basic composition of the two compounds, it is reasonable to believe that the differences observed between GC and PC could be attributable to the presence of graphene, which is able to affect the thermal stability of the material. Similar results, in terms of maximum degradation temperature and melting peaks, were found by Naskar et al. [24], who investigated the effects of waste plastic composed of HDPE e PP.

The differential thermal profiles (Figure 4b and Figure 5b) also allow the observation that the polymer crystalline parts of both compounds melt at temperatures (endothermic peaks) very close to the temperatures usually used for the production, laying, and compaction of the asphalt concretes, resulting in a potential worsening effect on the workability of the mixture produced with these binder blends.

### 3.3. Preparation and Aging of the Bituminous Blends

The two plastomeric compounds (PC and GC) were preliminarily grinded and then mixed with the plain bitumen to obtain the bituminous blends. The grinding process of the PC and GC pellets was carried out through a mechanical mill for three cycles. Then, the grinded compound was sieved in order to obtain the 0.50–1.00 mm fraction (Figure 1c,d), selected for the addition to the plain bitumen. A high shear mixer with a four-blade mixing head was used to mix the plain bitumen with 3% (by bitumen weight) of the grinded compound, at a temperature of 180 °C and a speed of 1000 rpm for 90 min. The percentage of 3% was selected by considering that these types of compounds are usually used for the dry modification of asphalt mixtures, with a percentage of 5% by bitumen weight according to the producer’s specification. As they are “rigid” and “not soft” materials, it can be assumed that only 60% of these compounds can actually react with the bitumen phase in the dry method. Thus, the value of 3% by bitumen weight was selected for the modification of the sole bitumen. In order to test the materials under the same condition and simulate the aging effect due to high temperature during the blending process, the plain bitumen (reference material) was similarly stirred by adopting the same protocol but without adding any compound. At the end of the blending phase, three materials were obtained: a plain bitumen, named PB, used as a reference, and two bitumen–compound blends, named PCB and GCB (prepared with PC and GC, respectively). For simplicity, the three materials are called blends in the text.

Both PCB and GCB were subjected to the “tube test” (EN 13399) [34], by carrying out two replicates for each material, to check the storage stability. Specifically, the bituminous blend was poured into an aluminum tube that was placed into an oven in a vertical position for 3 days at 180 °C. After removing the tube from the oven and left to cool down at room temperature, the tube was divided into three equal parts. In order to investigate the blend storage stability, the top and the bottom parts were compared in terms of viscosity, investigated at 160 °C with a roto-viscosimeter, and complex modulus, obtained through frequency sweep tests (frequency range from 0.159 to 15.9 Hz) at temperatures ranging from 4 to 88 °C. Two replicates for each blend were carried out.

The average results of viscosity tests (Figure 6a) showed that the bottom parts of both blends are characterized by significantly lower viscosity values than the top ones. Analogously, the top parts of GCB and PCB blends (Figure 6b) provide master curves (at 34 °C) significantly different from those of the corresponding bottom parts. Both results confirmed the expected lack of storage stability, as already found in other studies [26,27], given by the nature of the polymers used in the compounds (mainly polyethylene and polypropylene).

In order to reproduce the field conditions as accurately as possible, the produced blends were investigated under different aging conditions. Short- and long-term aging were carried out through the Rolling Thin Film Oven Test (EN 12607-1) [35] and Pressure Aging Vessel (EN 14769) [36], respectively. To avoid sample inhomogeneity due to the low storage stability of the blends investigated, each aging procedure was performed immediately after the blends‘ production without any storage. This means that the RTFOT was carried out immediately after the blending process and the PAV immediately after the RTFOT. Then, after the aging process, nine different blends were obtained, coded as shown in Table 3.

## 4. Testing Program and Protocols

The performance of each blend under different aging conditions was investigated by means of Fourier Transform Infrared Spectroscopy (FTIR), dynamic viscosity and frequency sweep tests, multiple stress creep recovery (MSCR) tests, and bitumen bond strength (BBS) tests on different aggregate types and surface conditions. In the following paragraphs, the testing procedures are described in detail, except the FTIR analysis that has already been described in Section 3.2.1.

### 4.1. Rheological Tests

The rheological characterization consisted of viscosity tests and dynamic shear rheometer (DSR) tests in different testing modes, i.e., frequency sweep and multiple stress creep recovery analysis.

The dynamic viscosity measurements were performed at three temperatures (115, 135, and 160 °C) by means of a Brookfield rotational viscosimeter compliant with the ASTM D4402 [37]. After filling the container, the blend was subjected to a 15 min conditioning time at each single temperature, followed by the measurement of the dynamic viscosity at a constant shear rate, corresponding to a fixed rotational viscosimeter working rate equal to 50%.

In order to analyze the time and temperature dependence of the investigated blends, frequency sweep tests were carried out according to EN 14770 [38]. Tests were performed in the temperature range from 4 to 88 °C with steps of 6 °C, by applying a testing frequency range between 0.159 and 15.9 Hz (0.1 and 100 rad/s, respectively) at each temperature. A plate–plate configuration was adopted with a diameter of 8 mm and a gap equal to 2 mm, from 4 to 34 °C, and a diameter of 25 mm and a gap equal to 1 mm, from 34 to 88 °C. A constant strain amplitude of 0.1% was maintained for each testing temperature and frequency.

The multiple stress creep recovery (MSCR) tests were carried out to evaluate the rutting behavior of the investigated blends, according to AASHTO T 350 [39]. Tests consisted of 10 creep–recovery cycles with a creep loading time of 1 s and a recovery time of 9 s for each testing temperature and applied load. Specifically, the blend behavior was analyzed by performing MSCR tests at four testing temperatures (58, 64, 70, and 76 °C) and two stress levels (0.1 and 3.2 kPa). Tests were conducted with DSR equipment in plate–plate geometry with a diameter of 25 mm and a gap equal to 1 mm on short-term aged blends.

### 4.2. Binder Bond Strength Test

The Binder Bond Strength (BBS) test was used to evaluate the adhesive/cohesive properties and moisture sensitivity of systems composed of the selected aggregate and the bituminous blend.

The BBS tests were performed according to AASHTO T 361 [40] by means of a modified Pneumatic Adhesion Tensile Testing Instrument (PATTI) [41], whose assembly is shown in Figure 7. The test consists of applying, through a pull-stub, a pulling force to a small binder specimen adhering onto a substrate, until failure occurs. At the end of the test, the failure pressure (i.e., the force needed to break the bond between the binder and the substrate) is recorded and converted into the pull-off tensile strength (POTS), used for characterizing adhesive/cohesive properties of the system.

The analysis of the BBS results also includes the assessment of the type of failure identified by visually inspecting the coating level of the contact area after the test. Specifically, the types of failure (Figure 8) can be divided into:(1)Cohesive (coded “C”), when failure occurs prevalently within the binder;(2)Adhesive (coded “A”), when failure occurs prevalently at the binder–substrate interface;(3)Hybrid (coded “C/A”), when failure occurs as a combination of adhesive and cohesive type.

The BBS test can be performed on any type of solid substrate characterized by different natures, as well as different surface properties. In this investigation, two types of mineral aggregate having different mineralogy were selected to produce suitable substrates, i.e., limestone (calcareous) and basalt (siliceous). Aggregate plates (10 × 10 × 1 cm^3^) were prepared by cutting quarry stone blocks to obtain a flat and horizontal surface. After that, two different surface conditions were considered for the aggregate plates before testing: virgin (uncoated surface) and pre-coated (coated surface with a bituminous film of SBS-modified binder). Under the pre-coated condition, the aggregate surface was prepared, as detailed by Canestrari et al. [42], in order to simulate an artificial Reclaimed Asphalt (RA) aggregate commonly used in road construction applications.

### 4.3. Testing Program

Preliminarily, the experimental program provided FTIR analysis on the three bituminous blends (PB, PCB, and GCB) under three different aging conditions (UN, ST, and LT), in order to evaluate the effect of compound addition and aging on the chemical structure of the plain bitumen. Two replicates were performed for each blend under each aging condition.

Afterward, the experimental program consisted of a rheological characterization, including dynamic viscosity tests with a rotational viscometer, as well as frequency sweep tests and multiple stress creep recovery tests using a DSR. Specifically, viscosity and DSR tests in frequency sweep mode were performed on the investigated blends under the three different aging conditions (UN, ST, and LT), whereas DSR tests in MSCR mode were only conducted on the short-term (ST) aged materials. At least two replicates were carried out for each testing temperature and aging condition. The testing program of each blend is summarized in Table 4.

As far as BBS tests are concerned, the experimental program provided 12 different binder–aggregate systems by combining two types of aggregate (basalt and limestone) under the two different surface conditions (virgin and pre-coated) with the three investigated blends (PB, PCB, and GCB). Moreover, for each blend–aggregate system, BBS tests were performed by considering a preliminary conditioning under dry or wet conditions. For the dry conditioning, the systems were left in a climate chamber at 25 °C for 24 h, whereas for the wet conditioning, the systems were put in a water bath at 40 °C for 24 h and were then further conditioned for 1 h in the climatic chamber at 25 °C. BBS tests were performed at 25 °C and 5 replicates were carried out for each system and conditioning type, for a total of 120 tests. Table 5 lists the testing parameters considered in the BBS test investigation.

## 5. Results and Discussions

### 5.1. Fourier Transform Infrared Spectroscopy

The comparison between the three blends (PB, PCB, and GCB) was carried out by considering the three aging conditions (UN, ST, and LT), as shown in Figure 9, where the FTIR spectra have been normalized with respect to the aromatics band and shifted along the y-axis. As the three blends provided similar spectra under UN conditions, as well as under ST and LT conditions, it seems that the plain bitumen is not chemically modified by the addition of each compound, leading to a dispersion of the compound in the bitumen, analogously to what happens, for instance, for mastics.

In order to better evaluate the influence of the aging process, a magnification of the range including the C=O and the aromatics bands (1760–1500 cm^−1^) is reported in the inset of Figure 9. Under UN and ST aged conditions, the three blends show very similar spectra, with a modest C=O band that appears around 1700 cm^−1^ in all three blends under the ST condition. On the contrary, in the long-term aged blends (LT), the band related to the absorption of C=O (responsible for the aging conditions) is more marked in the PB blend, followed by the PCB blend and then by the GCB blend. This means that the C=O and SO bands of PCB and GCB, compared to the bands of the reference material (PB), are influenced by the presence of compounds, especially GC, which limits its growth and thus the aging of the corresponding blend. These results seem to indicate that the compounds may be able to give slight protection to the bitumen against the aging, especially in the case of the GC compound.

### 5.2. Dynamic Viscosity Results

The dynamic viscosity results allow a comparison to be made to evaluate the effects of both the aging and presence of the compounds. The values of the logarithm of viscosity as a function of the test temperature for the three aging conditions are shown in Figure 10a, where only the PCB results have been reported given the analogies with the behavior also detected for the other two blends. The results confirmed that the presence of the compound does not influence the aging behavior of the PB, as the viscosity increases with the degree of aging, also for the blends PCB and GCB.

The influence of the presence of the compound under the same aging condition is shown in Figure 10b, where only the long-term aging condition has been reported, given the similarity with the behavior of the other two aging conditions. The analysis of the results showed that the PCB and GCB blends are more viscous than the PB one but are comparable to each other, allowing the conclusion to be made that the presence of graphene does not seem to cause substantial changes in terms of viscosity.

### 5.3. Frequency Sweep Test Results

Frequency sweep tests permitted the evaluation of the complex modulus (G*) and phase angle (δ) for each blend under each aging condition. The results reported in the black space (Figure 11) proved the general validity of the time–temperature superposition principle (TTSP), as continuous trends are obtained for all the blends. Moreover, under each aging condition, at high temperatures, the phase angle of the PB material approaches 90°, with the viscous component of the complex modulus (i.e., loss modulus G″) becoming predominant, contrarily to what happens for PCB and GCB blends, which show a reduction in the phase angle as the temperature increases, with a progressive predominancy of the elastic component of the complex modulus (i.e., storage modulus G′). A higher reduction in the phase angle is observed for the GCB blend with respect to the PCB blend, proving that the presence of graphene modifies the rheological behavior of the blend, especially at higher temperatures, as also confirmed by Figure 12 where the isochrones at 0.159 Hz of the complex modulus are shown. Indeed, at lower temperatures, the three blends provide analogous trends, which deviate when the temperature increases. Specifically, the GCB blend provides a significant reduction in the thermal sensitivity by showing a plateau (complex modulus slightly dependent or not dependent on the temperature change), which is more evident under the unaged condition with respect to the short- and long-term ones.

The master curves at 34 °C as a reference temperature of all the testing conditions are shown in Figure 13. Specifically, the master curves were obtained by applying appropriate shift factors, calculated using the closed form shifting (CFS) algorithm [43], able to eliminate the subjective uncertainties related to the manual shifting of the experimental data.

As can be noticed, at high and intermediate frequencies (i.e., low and intermediate temperatures), the three blends provide similar trends as, under this condition, it is likely that the characteristics of the bituminous matrix prevail with respect to the dispersed compound network. Contrarily, at low frequencies (i.e., high temperatures), the GCB blend is stiffer than the PCB one, which, in turn, is stiffer than the PB, meaning that the addition of the polymeric compound leads to an improvement of the rheological response at lower frequencies and/or higher temperature of the blends. In particular, the complex modulus trend of GCB and, to a lower extent, PCB as well, show a plateau at higher temperatures, typical of composite materials, highlighting that the solid plastic particles dominate the rheological behavior. The different behavior observed between the two blends (PCB and GCB) is likely caused by the combination of several factors such as the different types of plastics used to produce compounds, the presence of graphene, as well as the capacity of dispersion of the two different compounds within the plain bitumen. However, the differences among the investigated blends tend to reduce with the degree of aging (Figure 13).

Frequency sweep results were also employed to study the ability of PCB and GCB blends to resist rutting and fatigue cracking, by investigating the G*/sinδ and G*∙sinδ parameters, as specified by the AASHTO M 320-17 [44]. Specifically, bitumen is able to contrast fatigue cracking if G*∙sinδ ≤ 5000 kPa under the bitumen long-term aged condition, whereas it is able to contrast rutting if the ratio G*/sinδ is ≥ 1.00 kPa under the bitumen unaged condition and ≥ 2.20 kPa under the bitumen short-term aged condition.

Table 6 summarizes the temperatures at which these limits are respected for the three blends. The results confirmed that the blends are characterized by similar behavior at intermediate temperatures as they provide the same temperature limit when fatigue cracking is considered. On the contrary, the PCB and, even more, GCB blend show higher temperatures with respect to the PB bitumen, denoting an important contribution in rutting performance. However, as PCB and GCB are blends produced with modifying agents, multiple stress creep recovery (MSCR) tests are more suitable to investigate the rutting behavior [45], as the reversible conditions investigated in the frequency sweep tests are not appropriate for the estimation of the ability of the material to recover after the load removal.

### 5.4. Multiple Stress Creep Recovery Test Results

The MSCR test aims to quantify the nonlinear behavior of bituminous binders in order to predict their rutting response, overcoming the limitations of the traditional parameter (G*/sinδ) that is proven to underestimate the rutting resistance, especially for modified bitumens [45,46]. Specifically, the MSCR test allows the determination of the nonrecoverable creep compliance Jnr (calculated as the ratio between the accumulated strain and the applied stress), which is a good indicator of the resistance to the accumulation of permanent deformations as a function of the applied stress. In this investigation, the MSCR tests were conducted on the short-term aged blends according to the procedure described in the AASTHO T 350 Standard [39], at four temperatures: 58, 64, 70, and 76 °C. The test protocol consists of the application, in succession, of two constant stress creep phases (0.1 kPa and 3.2 kPa) of 1 s duration followed by a zero-stress recovery of 9 s duration, for 20 and 10 cycles, respectively. For each stress level, the average Jnr can be calculated as the mean of the values measured through the last 10 creep–recovery cycles.

The trends of Jnr0.1 and Jnr3.2 as a function of the temperature of the three blends (Figure 14) show that the PB material is more sensitive to permanent deformation than the PCB and GCB blends because of the higher Jnr values over the entire testing temperature range. This experimental finding clearly highlights the relevant contribution of the compound addition in terms of the reduction in permanent deformations, by revealing a similar anti-rutting potential for both PCB and GCB blends, as demonstrated by the comparable trend of the compliance curves at the applied stresses. Moreover, the comparison between Figure 14a,b shows that all the blends seem to have similar temperature sensitivity, as well as stress sensitivity, as no significant difference in the Jnr trends due to the temperature or applied stress increase is observed regardless of blend type.

In order to evaluate the capability of the investigated blends to recover deformation, MSCR results were also analyzed in terms of average percent recovery parameters R0.1 and R3.2 that are related to the stress levels of 0.1 kPa and 3.2 kPa, respectively. The results (Figure 15) showed that the PCB and GCB blends are characterized by a more elastic response as compared to the PB, indicating that the presence of the polymeric compound gives to the material a higher recovery ability after the loading phase, which is consistent with the increase in the elastic component of the complex modulus of PCB and GCB blends at the higher temperatures. As for blends, despite the PCB material seeming to recover more deformation than the GCB one at 64 and 70 °C, it is possible to state that both blends show a comparable temperature and stress dependency (decrease in R parameter with the increase in temperature and stress), highlighting similar effects due to the two types of compounds.

Finally, within the performance-grade designation of bituminous binders using the MSCR test (AASHTO M 332-18 [47]), the parameter J_nr_ and the percent difference in nonrecoverable creep compliance J_nr,diff_ (evaluated by comparing the J_nr_ measured at two different stress levels) were used to define the expected “traffic category” that the material under study could tolerate in relation to its rutting response. Based on the data collected during the MSCR tests at the temperature of 70 °C (selected as a function of the G*/sinδ analysis), the “traffic categories” that can be associated with the tested blends were obtained according to the AASHTO M 332–18 standard and are reported in Table 7. The results showed that the enhanced rutting behavior of both blends with the compounds (PCB and GCB) allows one to label them with a traffic category “V,” meaning that these blends are likely able to withstand a high traffic level (or standing traffic) without showing rutting issues, contrarily to the plain bitumen PB, which can only be used in the presence of standard traffic (or fast-moving traffic).

### 5.5. Binder Bond Strength Test

BBS tests were conducted to evaluate the bond strength between the investigated blends and the selected substrate. Specifically, the results are presented separately for substrates consisting of virgin aggregates or pre-coated aggregates.

#### 5.5.1. Virgin Aggregate Substrate

The results of BBS tests under dry and wet conditions, for both virgin limestone and basalt aggregates, are shown in Table 8.

Under dry conditions, all the blends tested with virgin limestone aggregates provide similar average POTS values and cohesive failures, allowing the observation that the presence of compounds does not seem to significantly affect the blend–limestone adhesion, as well as the inner cohesion of the blends. Under wet conditions, a decrease in the POTS values with respect to the dry condition is observed equal to about 20% for the PB material and about 6% for the PCB and GCB blends. Thus, in the case of limestone aggregate, the water conditioning seems to cause a reduction in the cohesive properties especially for the plain bitumen (PB), contrarily to what happens for PCB and GCB blends that do not experience any water effect. Moreover, all the failures are cohesive, denoting a good adhesive response in presence of moisture also.

The comparison between the three blends tested with virgin basalt aggregates (Table 8) shows that under dry conditions, all the systems are characterized by cohesive failures and provide similar average POTS values, which are consistent with those measured for the limestone aggregate systems. Regarding wet conditions, all the failures are adhesive-type and are combined with a marked reduction in the average POTS values of about 70% for the PB and the PCB blends and about 60% for the GCB blend. The occurrence of adhesive failures after immersion reveals that water, reaching the bonding layer interface, likely reduced the adhesion strength, which dropped below the cohesion strength of the binder (in accordance with the basically hydrophilic behavior attributable to the acid properties of siliceous minerals in the presence of water).

#### 5.5.2. Pre-Coated Aggregates Substrate

Table 9 shows the comparison between the blends tested under dry and wet conditions for the pre-coated limestone and basalt aggregates.

Regarding dry conditions, the three blends provide similar average POTS values regardless of aggregate type, with only cohesive failures, thus denoting similar cohesive properties consistently with those observed for the virgin aggregate systems. Regarding wet conditions, in both limestone and basalt pre-coated systems, the three blends show average POTS values that are comparable to each other and are not significantly affected by the wet conditioning, with differences between dry and wet conditions no higher than 20%. In addition, the failures are mostly cohesive-type regardless of substrate, except for the system GCB/basalt that shows a hybrid failure. The overall result analysis enables one to state that, in the case of the pre-coated aggregate, the presence of the compounds dispersed within the blends PCB and GCB does not seem to affect the adhesion, as well as the cohesion of the blend/aggregate systems under both dry and wet conditions. Moreover, the prevailing cohesive failure proves the good affinity between the investigated blends and the bituminous film that coats the aggregate, which, in turn, also ensures an efficient bond with the aggregate substrate after water conditioning.

#### 5.5.3. Comparison between Systems with Virgin and Pre-Coated Aggregate

The comparison between the systems with virgin and pre-coated aggregates for all the investigated blends under both testing conditions (dry and wet) can be performed by comparing Table 8 and Table 9.

The result analysis showed that for all the systems with pre-coated aggregate, the immersion in water does not produce any change in failure type, which is mostly cohesive (Table 9), contrarily to what happens to the systems with virgin aggregate, where the failure of basalt aggregate under wet conditions is always adhesive-type and associated with low POTS values (Table 8). Hence, the presence of the bituminous coating on the aggregate surface, simulating a reclaimed asphalt particle, does not lead to any significant reduction in the average POTS value (which even seems to increase when limestone aggregates are considered). This finding allows the assertion to be made that pre-coated aggregates improve the performance in terms of adhesion with the blend, thus reducing the moisture sensitivity of the system, as already found by Canestrari et al. [42]. This is mainly due to the development of physicochemical interactions between the fresh bitumen of the blend and the bituminous film coating the aggregate, which also ensures a more efficient bond with the aggregate substrate after water conditioning. Indeed, such an improved interaction can be partly explained considering that the aging experienced by the thin bitumen film results in an increase in adhesiveness properties of bitumen [48], which likely enhances the bond at the interface, making the system more resistant to stripping.

Finally, regarding the dry condition, it can be noticed that all the failures are cohesive-type, meaning that no systems are penalized in terms of loss in adhesion, despite the systems with virgin aggregate showing average POTS values that tend to be higher than those of the systems with pre-coated substrates. This finding may be due to a different physicochemical interaction of the blend with the coated aggregate instead of the surface of the virgin aggregate, which results in a slightly lower inner cohesion (i.e., POTS) of the specimen.

## 6. Conclusions

In this study, two different bituminous blends (PCB and GCB) obtained by adding a plain bitumen (PB) with two plastomeric compounds, one of which contains recycled plastics and graphene, used for asphalt concrete dry modification, were investigated. As it is well known that the rheological characteristics of an asphalt mixture strongly depend on the properties of the constituting bitumen, the focus was on studying the effects of the compounds on the rheological response of the binder phase, as well as the adhesion properties, in comparison with a reference plain bitumen (PB). To this end, an extensive laboratory characterization consisting of dynamic viscosity tests, frequency sweep tests, multiple stress creep recovery (MSCR), as well as bitumen bond strength (BBS) tests was performed on the selected binders.

From the experimental results, the following conclusions can be drawn:(1)Preliminary chemical investigations showed that both compounds are basically composed of polyethylene and polypropylene. The compound named GC, including a small amount of graphene, appears to affect the bitumen aging.(2)The results of rheological tests carried out on samples subjected to the Tube Test confirmed the lack of storage stability for the PCB and GCB blends, similarly to what happens for most of the modified asphalt binders.(3)In general, PCB and GCB blends showed a higher stiffness (complex modulus) at low loading frequencies and/or high temperatures, compared to PB bitumen, highlighting that the solid plastic particles dominate the rheological behavior. This result was also confirmed by the enhanced rutting behavior of both blends combined to a more elastic response as compared to the PB bitumen.(4)The inclusion of compounds within the blends PCB and GCB does not penalize the bond strength of the bitumen–aggregate system, regardless of the type of aggregate (limestone or basalt) and the surface condition of the substrate of the aggregate (virgin or pre-coated).(5)The overall analysis of the results allows us to state that the performance of PCB and GCB blends is comparable, highlighting that the use of recycled plastics (GCB material) is promising for the modification of bituminous materials.

In conclusion, as these compounds are designed for the proper modification of asphalt concrete through a dry process, it is worth highlighting that their effect on the asphalt mixture behavior is currently in progress as part of a wider research investigation focusing on the study of plastics from recycled sources.

## Figures and Tables

**Figure 1 materials-15-00516-f001:**
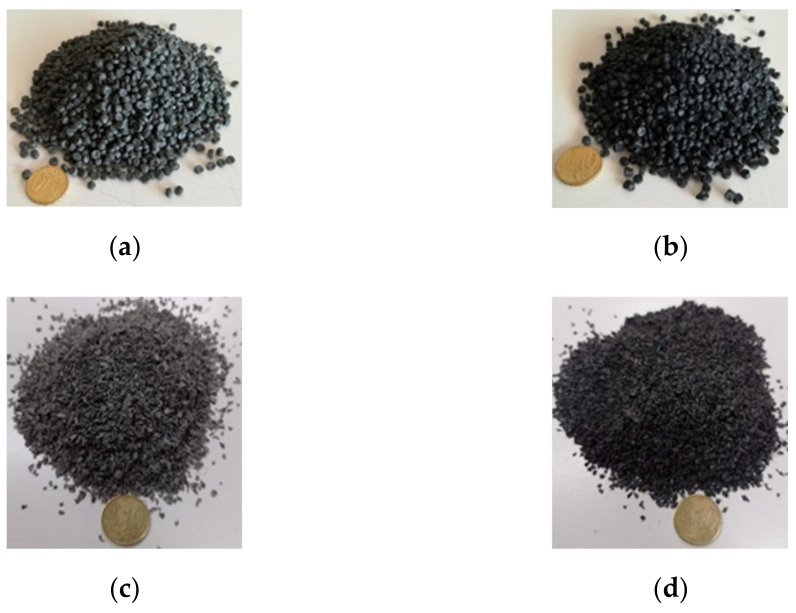
Plastomeric compounds: (**a**) PC-pellets; (**b**) GC-pellets. Fraction 0.50–1.00 mm after grinding and sieving: (**c**) PC; (**d**) GC.

**Figure 2 materials-15-00516-f002:**
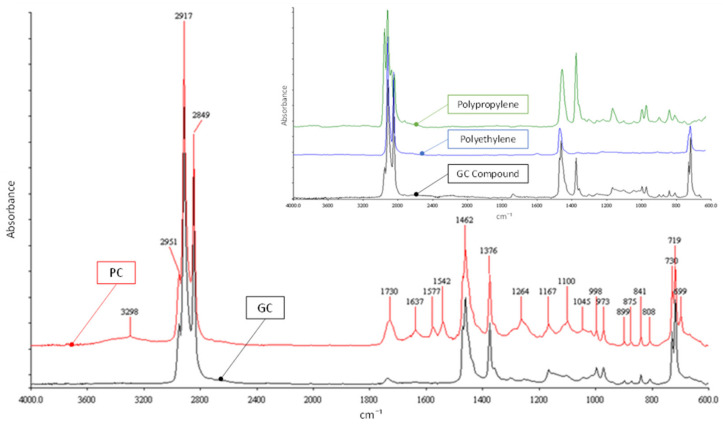
FTIR spectra of the compounds: GC (black line) and PC (red line). In inset: the comparison between GC (black line) and data available in the literature, related to PE (blue line) and PP (green line).

**Figure 3 materials-15-00516-f003:**
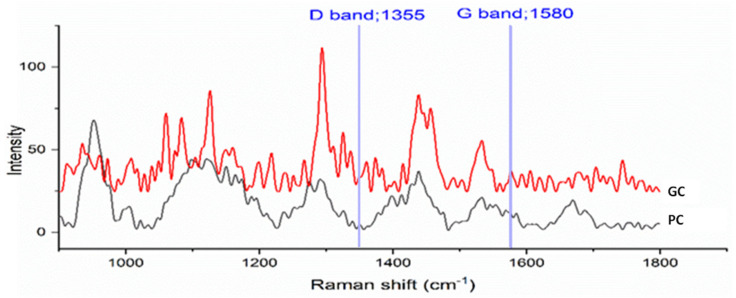
Raman spectra representative of the GC (red line) and PC (black line) compounds.

**Figure 4 materials-15-00516-f004:**
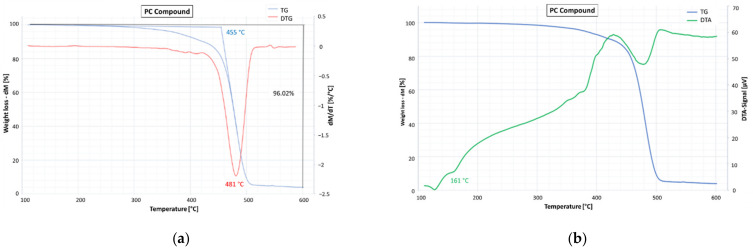
PC compound: (**a**) percentage mass loss TG (blue line) and first derivate of mass loss DTG (red line); (**b**) TG (blue line) and differential thermal profile DTA (green line).

**Figure 5 materials-15-00516-f005:**
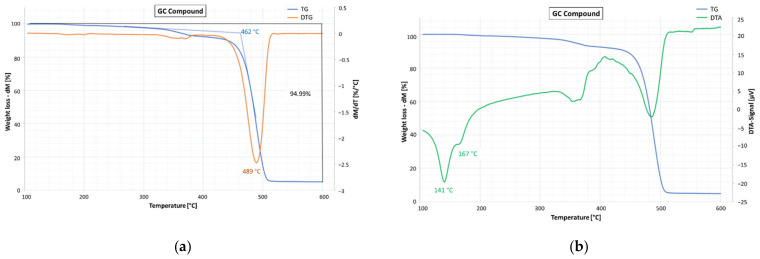
GC compound: (**a**) percentage mass loss TG (blue line) and first derivative of mass loss DTG (red line); (**b**) TG (blue line) and differential thermal profile DTA (green line).

**Figure 6 materials-15-00516-f006:**
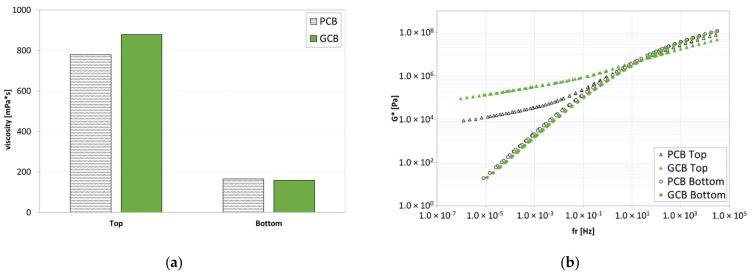
Analysis after tube test: (**a**) viscosity at 160 °C; (**b**) master curve at 34 °C.

**Figure 7 materials-15-00516-f007:**
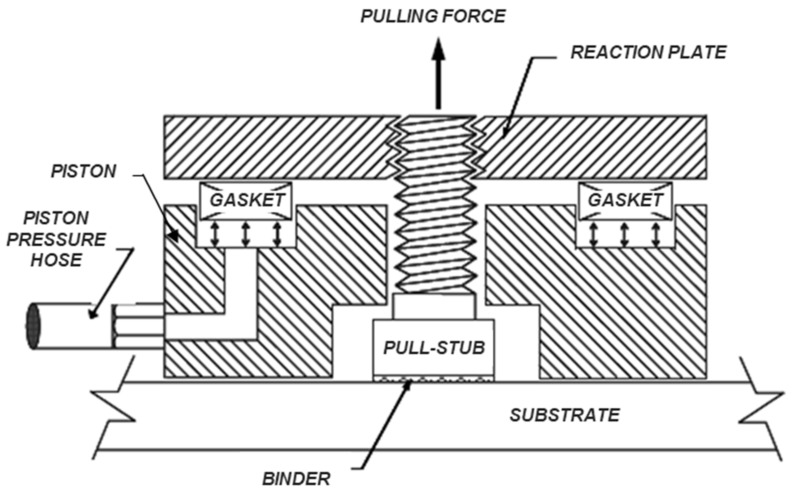
Schematic cross-section of BBS test device.

**Figure 8 materials-15-00516-f008:**
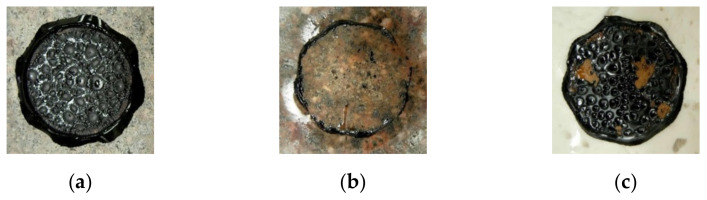
Examples of failure type: (**a**) cohesive; (**b**) adhesive; (**c**) hybrid.

**Figure 9 materials-15-00516-f009:**
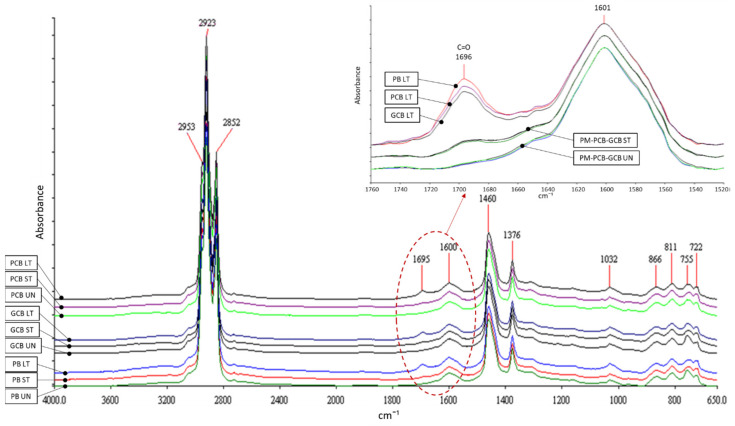
Average FTIR spectra of bituminous blends under different aging conditions. Inset shows the magnification of the FTIR spectra in the spectral range 1760–1500 cm^−1^.

**Figure 10 materials-15-00516-f010:**
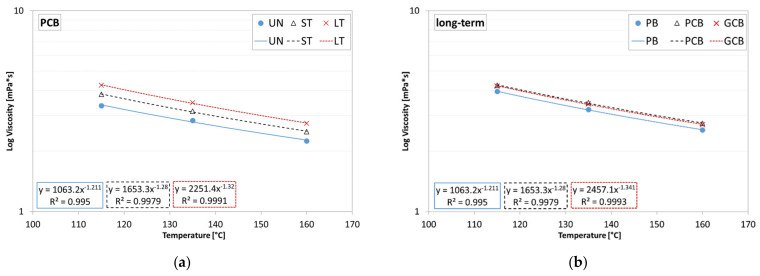
Average results of viscosity: (**a**) different aging conditions of blend PCB; (**b**) different blends under long-term aging conditions.

**Figure 11 materials-15-00516-f011:**
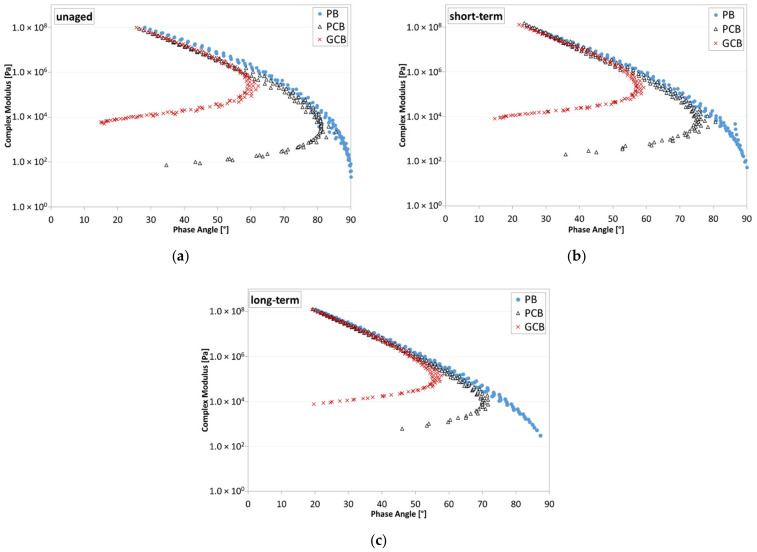
Black space of bituminous blends: (**a**) unaged condition; (**b**) short-term aged condition; (**c**) long-term aged condition.

**Figure 12 materials-15-00516-f012:**
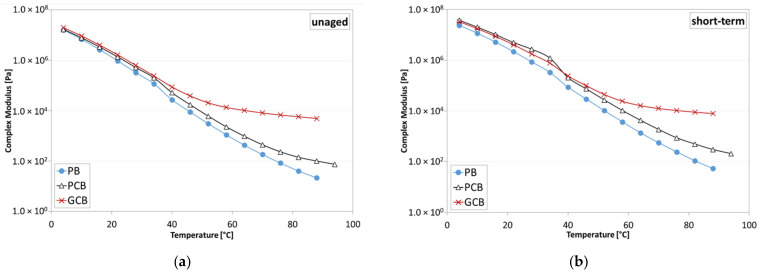
Isochrones of the complex modulus at 0.159 Hz: (**a**) unaged condition; (**b**) short-term aged condition; (**c**) long-term aged condition.

**Figure 13 materials-15-00516-f013:**
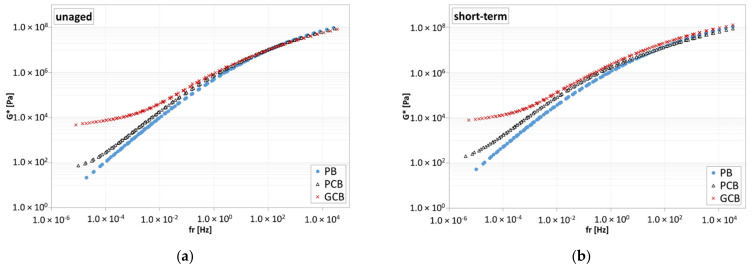
Master curves of bituminous blends: (**a**) unaged condition; (**b**) short-term aging condition; (**c**) long-term aging condition.

**Figure 14 materials-15-00516-f014:**
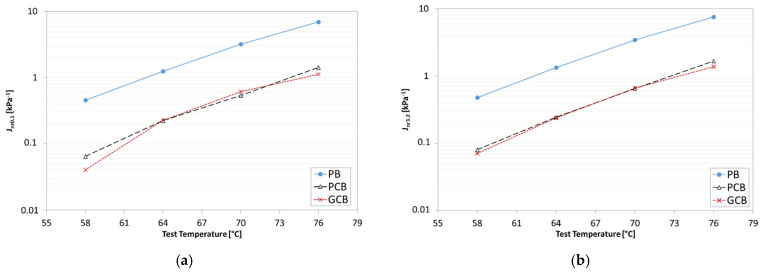
Nonrecoverable creep compliance of the three blends: (**a**) J_nr0.1_; (**b**) J_nr3.2_.

**Figure 15 materials-15-00516-f015:**
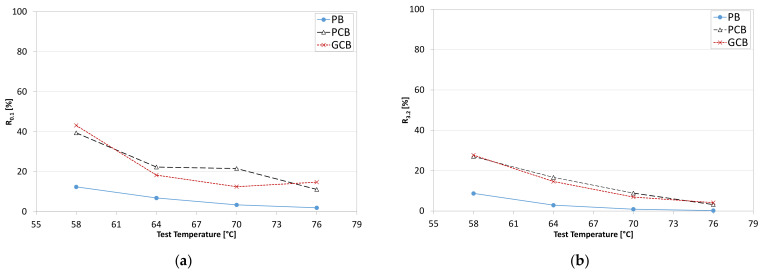
Average percent recovery R of the three blends: (**a**) at 0.1 kPa; (**b**) at 3.2 kPa.

**Table 1 materials-15-00516-t001:** Main characteristics of the reference plain bitumen.

Characteristics	Standard	Unit	Performance
Penetration at 25 °C	EN 1426	0.1mm	51.5
Softening point	EN 1427	°C	49.7
Retained penetration	EN 1426	%	77
Increase in softening point after RTFOT	EN 1427	°C	4.6
Mass loss (75 min at 163 °C)	EN 12607/1	%	≤ 1

**Table 2 materials-15-00516-t002:** Main properties of the compounds.

Property	Unit	Compound	
		PC	GC
Apparent density at 25 °C	g/cm^3^	0.4–0.6	0.4–0.6
Softening point	°C	160–180	160–180
Fluidity index at 190 °C/5 kg	g/10 min	-	4–10

**Table 3 materials-15-00516-t003:** Code of the blends.

Blend	Aging Condition		
	Unaged	Short-Term Aged	Long-Term Aged
PB	PB_UN	PB_ST	PB_LT
PCB	PCB_UN	PCB_ST	PCB_LT
GCB	GCB_UN	GCB_ST	GCB_LT

**Table 4 materials-15-00516-t004:** Experimental program of each blend.

Test	Aging Conditions	Test Parameters
FTIR analysis	UN, ST, LT	Room temperature
Viscosity test	UN, ST, LT	Temperature: 115, 135, 160 °C
Frequency sweep test	UN, ST, LT	Temperature: from 4 °C to 88 °C (by step of 6 °C) Frequency: form 0.159 Hz to 15.9 Hz Strain amplitude = 0.1%
MSCR test	ST	Temperature: 58, 64, 70, and 76 °C Stress levels: 0.1 and 3.2 kPa

**Table 5 materials-15-00516-t005:** BBS testing conditions.

Parameter	Testing Conditions
Blend	PB, PCB, GCB
Aggregate type	Limestone, basalt
Surface condition	Uncoated, coated
Conditioning type	Dry, wet

**Table 6 materials-15-00516-t006:** Temperature limits at which SHRP requirements are met.

Binder	G*∙sinδ	G*/sinδ
PB	34 °C	70 °C
PCB	34 °C	76 °C
GCB	34 °C	88 °C

**Table 7 materials-15-00516-t007:** Traffic categories in accordance with AASHTO M 332-18 Standard.

Blend	J_nr3.2_ [kPa^−1^]	J_nr,diff_ [%]	Traffic Category
PB	3.44	8.23	S—Standard Traffic
PCB	0.65	21.36	V—Very High Traffic
GCB	0.66	7.75	V—Very High Traffic

**Table 8 materials-15-00516-t008:** BBS testing results for systems with virgin aggregate.

Binder	Virgin Aggregate	Conditioning DRY			Conditioning WET		
		POTS (kPa)		Failure Type	POTS (kPa)		Failure Type
		Average	Std. Dev.		Average	Std. Dev.	
PB	Limestone	3374	192	C	2707	358	C
	Basalt	3290	145	C	973	80	A
PCB	Limestone	3136	183	C	2956	92	C/A
	Basalt	3465	204	C	1089	92	A
GCB	Limestone	3135	340	C	2999	286	C
	Basalt	3288	846	C	1379	216	A

**Table 9 materials-15-00516-t009:** BBS testing results for systems with pre-coated aggregate.

Binder	Pre-Coated Aggregate	Conditioning DRY			Conditioning WET		
		POTS (kPa)		Failure Type	POTS (kPa)		Failure Type
		Average	Std. Dev.		Average	Std. Dev.	
PB	Limestone	2743	205	C	3363	224	C
	Basalt	2769	194	C	2403	220	C
PCB	Limestone	2376	235	C	3039	161	C
	Basalt	2746	150	C	2211	248	C
GCB	Limestone	2509	181	C	2691	223	C
	Basalt	2749	183	C	2486	202	C/A

## Data Availability

The data presented in this study are available on request from the corresponding author.

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
