# Peer review of "Asphalt Binder Modification with Plastomeric Compounds Containing Recycled Plastics and Graphene"

_materials, 2022, doi:10.3390/ma15020516_

Round 1

Reviewer 1 Report

The authors present an experimental investigation on bitumen added with two plastomeric compounds, containing recycled plastics and graphene to study the effects of the compounds on the rheological response of the binder phase as well the adhesion properties. The methodology was described comprehensively and the results are meaningful. However, there are some problems existing in this paper which the authors must pay attention to deal with.

  1. What does the symbol ‘÷’ mean in Table 1 and Table 2?
  2. Replace Figure 5 with a clear one.
  3. The GC addictive used in this manuscript is plastomeric compound containing recycled plastics and graphene. Is it a waste material recycled from industrial product or two waste materials (plastics and graphene) mixed together? Please clarify. Percentages of plastic and graphene in the GC addictive should also be given.
  4. Page 6, section 3.3. A percentage of 3%(by bitumen weight) of grinded compound was added in the plain bitumen. The reason of the addictive amount should be stated briefly.
  5. Page 10, section 5.1. It should provide more description and interpretation of the test results.

Reviewer 2 Report

Comments

Title: Asphalt binder modification with plastomeric compounds containing recycled plastics and graphene

Article number: materials-1530413

The subject of the paper is very important topic, and gives a new information about the used additives for bitumen.

The paper is well written. All the presented elements are presented in a smooth and clear manner.

I recommended it for publication after doing very minor comments. 

Lines

Comments

17

(The results indicate……….)

it is better to refer for which tests you mean, or …. rheological response indicates ….

101

(Both blends are particularly hard to touch………….)

what do you mean (hard to touch)?  … do you mean they have a rough texture?

185

187

Figure 4 and Figure 5 are not clear. Please replace the figures with clearest other.

342

In Figure 9: we cannot distinguish between blends. You need to mention about the blends in the main figure.

Or you need to separate the FTIR for the blends with their ageing stages.

Also, try to maximize the box to be more clear for your readers.

Reviewer 3 Report

  1. Introduction should highlight the novelty of this work.
  2. Page 3, table 1: It is necessary to add e the actual characteristics of bitumen classified as 50/70 pen grade.
  3. Page 3, Lines 98-104: Please explain more about Fig 1. How were the blends made?
  4. Page 3, figures 4 and 5: What different?
  5. Page 3, lines 189-203: Polymer content is not varied in this study, which does not enough to adequately decide the tasks.
  6. Pages 5-18: The author should compare and discuss their data based on the literature and results of their most relevant peers.
  7. Conclusions: It is not sufficiently clear what new has been installed.
